# Cutting Long Gradient Flows: Decoupling End-to-End Backpropagation Based on Supervised Contrastive Learning

## Abstract

End-to-end backpropagation (BP) is the foundation of current deep learning technology. Unfortunately, as a network becomes deeper, BP becomes inefficient for various reasons. This paper proposes a new methodology for decoupling BP to transform a long gradient flow into multiple short ones in order to address the optimization issues caused by long gradient flows. We report thorough experiments conducted to illustrate the effectiveness of our model compared with BP, Early Exit, and associated learning (AL), a state-of-the-art methodology for backpropagation decoupling. We release the experimental code for reproducibility.

## 1 Introduction

Current deep learning technology largely depends on backpropagation and gradient-based learning methods (e.g., gradient descent) for model training. Meanwhile, many successful applications rely on extremely deep neural networks; for example, Transformer contains at least 12 layers (most have several sublayers) (Vaswani et al., 2017), BERT has 12 to 24 layers (most also have several sublayers) (Devlin et al., 2018), and GoogLeNet has 22 layers (many layers are Inception modules containing several sublayers) (Szegedy et al., 2015). However, training a deep network based on backpropagation is inefficient for many reasons. First, a long gradient flow may suffer from gradient vanishing or explosion (Hochreiter, 1998). Second, a long gradient flow may lead to unstable gradients in the early layers (the layers close to the input layer) (Nielsen, 2015). Third, backpropagation results in backward locking, meaning that the gradient of a network parameter can be computed only when all other gradients on which it depends have been computed (Jaderberg et al., 2017). These issues may become severe bottlenecks, especially when a network is deep. To train deep networks more efficiently, researchers have developed various strategies, such as batch normalization, gradient clipping, new activation functions (e.g., ReLU and leaky ReLU), new network architectures (e.g., LSTM (Hochreiter & Schmidhuber, 1997)), and many more.

Since a long gradient flow is a root cause of the above issues, a possible way to eliminate these issues is to shorten the length of the gradient flow, for example, by cutting a network into multiple components and assigning a local objective to each component. In this way, a long gradient flow can be divided into multiple shorter pieces, which should, at least partially, address the issues of vanishing/exploding gradients, unstable gradients in early layers, and backward locking. Perhaps the most straightforward approach for assigning a local objective to a component is by adding a local auxiliary classifier that outputs a predicted $\hat{y}$ and updates the local parameters based on the difference between $\hat{y}$ and the ground-truth target $y$. We call this strategy "Early Exit" in this paper because each such auxiliary classifier can be regarded as an exit of the neural network. The concept of Early Exit has been used in many previous studies, e.g., Mostafa et al. (2018); Teerapittayanon et al. (2016); Szegedy et al. (2015). However, most of these studies have used Early Exit for other purposes, e.g., creating multiple prediction paths or helping to obtain gradients for the parameters in the early layers. Consequently, these studies have not investigated the separation of end-to-end backpropagation (BP) into multiple pieces, and the associated gradient flows are still long. In addition, even if Early Exit is used to isolate the gradient flow, as shown in (Mostafa et al., 2018), the test accuracies are lower than those of models trained via BP. There are other methods of cutting long gradient flows (Jaderberg et al., 2017; Czarnecki et al., 2017; Löwe et al., 2019; Wu et al., 2022; Kao & Chen, 2021). However, most of these methods have been applied only to simple networks, and

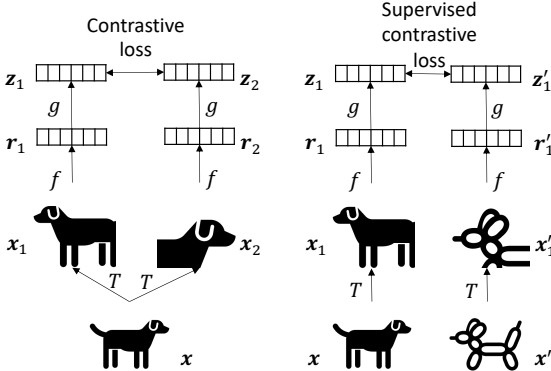

Figure 1: An illustration of contrastive learning and supervised contrastive learning.

their test accuracies are still unsatisfactory likely because the local objectives may not align with the global objective; details will be discussed in Section 4.

This paper proposes a new methodology, Delog-SCL, which decouples the long gradient flow of a deep neural network by leveraging supervised contrastive learning (SCL). In this design, the forward path transforms an input $x$ into the corresponding prediction $\hat{y}$ as usual. However, the gradient flow on the backward path is blocked between different components. Instead of using a global objective, we assign a local objective to each component and force each gradient flow to remain within one component. We present experiments conducted on multiple open datasets and compare the result with those of models trained via BP, Early Exit, and associated learning (AL) (Wu et al., 2022; Kao & Chen, 2021), a state-of-the-art methodology for BP decoupling that yields results comparable to those of BP. We find that Delog-SCL outperforms AL in terms of test accuracy in most cases with fewer parameters. Additionally, since our method has a more straightforward network architecture than AL, our method could be a favorable alternative to AL.

The rest of the paper is organized as follows. In Section 2, we introduce Delog-SCL and its properties. Section 3 presents a comparison of Delog-SCL with BP, Early Exit, and AL in terms of their test accuracies and numbers of parameters. We also report the results of analyses on certain properties of Delog-SCL in the same section. Section 4 reviews previous works on BP decoupling and presents a comparison of these works with our model. We conclude our contribution in Section 5.

## 2 METHODOLOGY

### 2.1 PRELIMINARIES: CONTRASTIVE LEARNING AND SUPERVISED CONTRASTIVE LEARNING

Contrastive learning (CL) is a self-supervised technique for learning visual representations of images. Referring to the left of Figure 1, given an image $x$, CL involves generating different views (i.e., $x_1$ and $x_2$ in Figure 1) via the same family of data augmentations $T$. The generated views ($x_1$ and $x_2$) are further transformed via an encoder function $f$ and a projection head $g$ to minimize the contrastive loss between the output vectors (i.e., $z_1$ and $z_2$). After training, the projection head $g$ is disregarded, and only the encoder $f$ is used to generate the visual representations of images (Chen et al., 2020). In other words, given an anchor image $x$, CL relies on regarding its augmented images as positive instances and all other images as negative instances and considering that positive pairs should be close after encoding and projection.

SCL refers to the extension of CL from a self-supervised setting to a fully supervised setting. Therefore, the training data for SCL consist of not only the training images themselves but also the classes of those images. Referring to the right of Figure 1, given an anchor image $x$ of class $c$, the positive instances include the other images of class $c$ in the same batch, whereas all other images in the same batch are regarded as negative instances (Khosla et al., 2020).

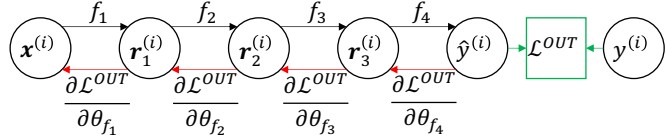

Figure 2: An example neural network with 3 hidden layers. The black arrows correspond to the forward path, the red arrows correspond to the backward path, and the green box denotes the comparison of the distance between two incoming variables $\hat{y}^{(i)}$ and $y^{(i)}$.

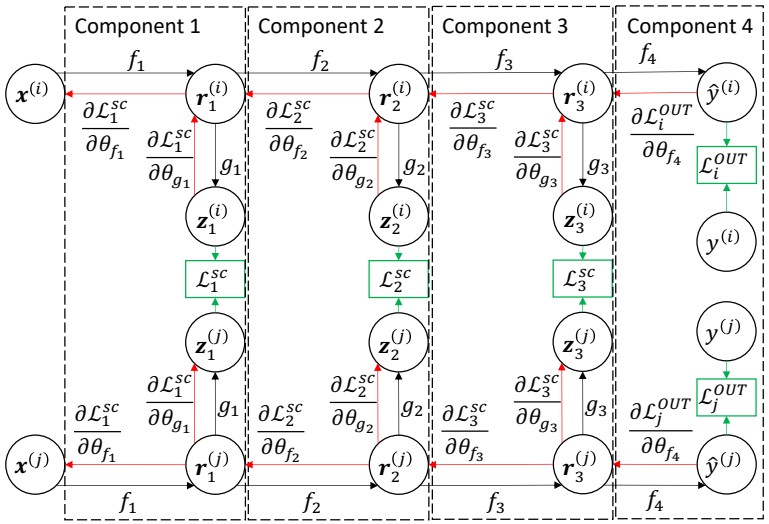

Figure 3: An example of a decoupled neural network based on supervised contrastive learning. The black arrows correspond to the forward path, the red arrows correspond to the backward path, and the green box denotes the comparison of the distance between two incoming variables.

## 2.2 DECOUPLING END-TO-END BACKPROPAGATION VIA SUPERVISED CONTRASTIVE LEARNING

This section presents Delog-SCL, which leverages the supervised contrastive loss to split one long gradient flow in a deep neural network into multiple shorter ones.

Let us first consider a standard neural network with 3 hidden layers as an example. As shown in Figure 2, $\boldsymbol{x}^{(i)}$ refers to an input image $i$, and the function $f_\ell$ ($\ell = 1, \ldots, 4$) transforms $\boldsymbol{r}_{\ell-1}^{(i)}$ into $\boldsymbol{r}_\ell^{(i)}$ (under the assumptions that $\boldsymbol{x}^{(i)} = \boldsymbol{r}_0^{(i)}$ and the predicted class $\hat{y}^{(i)} = \boldsymbol{r}_4^{(i)}$). Depending on the network architecture, the functions $f_\ell$ could be various well-known neural network layers, such as fully connected layers, convolutional layers, pooling layers, or residual blocks. The objective $\mathcal{L}^{OUT}$ is determined by the task type. For example, if we are addressing a classification task, we could use the cross-entropy loss between the predicted class $\hat{y}^{(i)}$ and the ground-truth class $y^{(i)}$ as the objective $\mathcal{L}^{OUT}$. We use backpropagation to obtain $\partial\mathcal{L}^{OUT}/\partial\theta_{f_\ell}$ for each layer $\ell$, where $\theta_{f_\ell}$ represents the parameters of function $f_\ell$. Once the gradients are obtained, we can use gradient-based optimization strategies, e.g., gradient descent, to update the parameter values. Given a neural network with $H$ hidden layers, it can be seen that the longest gradient flow is constructed as a product of $H + 2$ local gradients. For example, to obtain $\partial\mathcal{L}^{OUT}/\theta_{f_1}$ in a network with 3 hidden layers (as shown in Figure 2), we need the following:

$$\frac{\partial\mathcal{L}^{OUT}}{\partial\theta_{f_1}} = \frac{\partial\mathcal{L}^{OUT}}{\partial\hat{y}^{(i)}} \times \frac{\partial\hat{y}^{(i)}}{\partial r_3^{(i)}} \times \frac{\partial r_3^{(i)}}{\partial r_2^{(i)}} \times \frac{\partial r_2^{(i)}}{\partial r_1^{(i)}} \times \frac{\partial r_1^{(i)}}{\partial\theta_{f_1}}. \tag{1}$$

The number of terms of this product grows linearly with the depth of the network. Therefore, as networks become deeper, their long gradient flows cause several optimization issues, as discussed in Section 1.

We use Figure 3 to illustrate our strategy of cutting a long gradient flow into several local gradients for a neural network with 3 hidden layers. Let $r_0^{(i)}$ (i.e., $x^{(i)}$) and $r_0^{(j)}$ (i.e., $x^{(j)}$) be two image views in the same batch ($r_0^{(i)}$ and $r_0^{(j)}$ may or may not be augmented images, i.e., views, from the same image). We use $f_1$ to transform each of them, obtaining $r_1^{(i)}$ and $r_1^{(j)}$, and further use the function $g_1$ to convert them into $z_1^{(i)}$ and $z_1^{(j)}$, respectively. The functions $f_1$ and $g_1$ can be regarded as the encoding function and the projection head, respectively, in CL (refer to Figure 1). We repeat the same process for each hidden layer $\ell$ to form the corresponding component $\ell$. If $x^{(i)}$ and $x^{(j)}$ are two different views of the same image or if $y^{(i)}$ (the class of $x^{(i)}$) is equal to $y^{(j)}$ (the class of $x^{(j)}$), then we should ensure that $z_\ell^{(i)}$ is close to $z_\ell^{(j)}$ for all $\ell$. Otherwise, we should increase the distance between $z_\ell^{(i)}$ and $z_\ell^{(j)}$. In the last layer, we compute the distance between $\hat{y}^{(i)}$ and $y^{(i)}$ as the loss $\mathcal{L}_i^{OUT}$. Eventually, we define the local supervised contrastive loss $L_\ell^{SC}$ for batch $B$ in layer $\ell$ as shown in Equation 2:

$$\mathcal{L}_\ell^{SC} = \sum_{\forall i \in B} \frac{-1}{|P(i)|} \sum_{\forall p \in P(i)} \log \frac{\exp\left(z_\ell^{(i)} \cdot z_\ell^{(p)}/\tau\right)}{\sum_{\forall j \in B} I(j \neq i) \exp\left(z_\ell^{(i)} \cdot z_\ell^{(j)}/\tau\right)}, \tag{2}$$

where $B = 1, 2, \ldots, N$ represents a batch of multiview images, $P(i)$ is the set of all positive samples for an image $i$, $\tau$ is a hyperparameter, and $I(j \neq i) \in \{0, 1\}$ is an indicator function that returns 1 if $j \neq i$ and 0 otherwise.

Ultimately, the global objective function is an accumulation of the local supervised contrastive losses and the losses in the output layer, as defined below:

$$\mathcal{L} = \sum_{\ell=1}^{H} \mathcal{L}_\ell^{SC} + \sum_{i=1}^{N} \mathcal{L}_i^{OUT}, \tag{3}$$

where $H$ is the number of hidden layers and $\mathcal{L}_i^{OUT}$ is the $i$th loss in the output layer (refer to Figure 3).

The computation of $\mathcal{L}_\ell^{SC}$ and the pseudo code of Delog-SCL for a 3-layer vanilla ConvNet is given in Algorithm 1 and Algorithm 2 in Appendix A.5.

### 2.3 FORWARD PATH, BACKWARD PATH, AND INFERENCE FUNCTION

For a regular neural network (e.g., Figure 2), the forward path and the inference function are identical, and the backward path is simply obtained by inverting the direction of the forward path. However, the situation is more complicated in our case because we divide the global objective into several local ones. Consequently, we have multiple short forward paths, multiple short backward paths, and one inference path. Thus, the inference path and the forward paths are no longer identical in Delog-SCL.

During training, each component $\ell$ has its own forward and backward paths. Taking Figure 3 as an example, the forward path of component $\ell$ transforms each $r_{\ell-1}^{(i)}$ into $r_\ell^{(i)}$ via the local encoding function $f_\ell$ and further transforms each $r_\ell^{(i)}$ into $z_\ell^{(i)}$ via the local projection head $g_\ell$. On the backward path, each hidden layer computes $\partial\mathcal{L}_\ell^{SC}/\partial\theta_{g_\ell}$ and $\partial\mathcal{L}_\ell^{SC}/\partial\theta_{f_\ell}$ based on the chain rule and updates the parameters by means of gradient-based optimization strategies. We block the gradient flow between each component.[1] As a result, each gradient flow remains within one component and is therefore short. Equation 4 and Equation 5 show these local gradient flows.

---

[1]The gradient flow can be blocked by using `Tensor.detach()` in PyTorch or `tf.stop_gradient` in TensorFlow.

$$\frac{\partial \mathcal{L}_\ell^{SC}}{\partial \theta_{g_\ell}} = \frac{\partial \mathcal{L}_\ell^{SC}}{\partial \boldsymbol{z}_\ell^{(i)}} \times \frac{\partial \boldsymbol{z}_\ell^{(i)}}{\partial \theta_{g_\ell}}. \tag{4}$$

$$\frac{\partial \mathcal{L}_\ell^{SC}}{\partial \theta_{f_\ell}} = \frac{\partial \mathcal{L}_\ell^{SC}}{\partial \boldsymbol{z}_\ell^{(i)}} \times \frac{\partial \boldsymbol{z}_\ell^{(i)}}{\partial \boldsymbol{r}_\ell^{(i)}} \times \frac{\partial \boldsymbol{r}_\ell^{(i)}}{\partial \theta_{f_\ell}}. \tag{5}$$

Eventually, even if we construct a deep neural network, the cost of computing each $\partial \mathcal{L}^{SC}/\partial \theta_{f_\ell}$ and each $\partial \mathcal{L}^{SC}/\partial \theta_{g_\ell}$ remains constant. Additionally, the gradient flow in the output layer is also short: we simply compute $\partial \mathcal{L}_k^{out}/\partial \theta_{f_{H+1}}$ (where $H$ is the number of hidden layers). This design alleviates various issues caused by long gradient flows.

In the inference (prediction) phase, we need the functions $f_\ell$ but not $g_\ell$, as shown by Equation 6:

$$\hat{y}^{(i)} = f_{H+1} \circ f_H \circ \ldots \circ f_2 \circ f_1(\boldsymbol{x}^{(i)}), \tag{6}$$

where $\circ$ is the function composition operator ($H = 3$ for the example illustrated in Figure 3).

Although our proposed method (e.g., Figure 3) involves more parameters than a standard neural network structure (e.g., Figure 2) during training, they have the same number of parameters during inference because both of them use only the functions $f_\ell$. Therefore, they have the same hypothesis space. The parameters that participate in the inference phase (denoted by $\theta_{f_\ell}$) are called the *effective parameters*, and the parameters used during training but not during inference (denoted by $\theta_{g_\ell}$) are called the *affiliated parameters*.

## 2.4 PROPERTIES

Table 1: An illustration of the training process pipeline

|       | $t_1$  | $t_2$  | $t_3$  | $t_4$  | $t_5$  | $t_6$  | ... |
|-------|--------|--------|--------|--------|--------|--------|-----|
| $B_1$ | Task 1 | Task 2 | Task 3 |        |        |        |     |
| $B_2$ |        | Task 1 | Task 2 | Task 3 |        |        |     |
| $B_3$ |        |        | Task 1 | Task 2 | Task 3 |        |     |
| $B_4$ |        |        |        | Task 1 | Task 2 | Task 3 |     |
| ...   |        |        |        |        |        |        |     |

In this section, we discuss three properties of our proposed model — short gradient flows, a flexible structure, and the ability to perform parallel (pipelined) training.

As discussed in Section 2.3, training a regular neural network with BP requires a gradient flow of length $O(H)$. In contrast, the length of each gradient flow in our model is independent of the number of layers; the length is always a constant. Therefore, the various optimization issues resulting from long gradient flows, as discussed in Section 1, are eliminated (or at least alleviated).

The network structure is more flexible and perhaps easier to understand than that of associated learning (AL), a state-of-the-art methodology for decoupling BP in terms of test accuracy (Kao & Chen, 2021; Wu et al., 2022). Specifically, AL involves projecting the features $\boldsymbol{x}$ and the target $y$ into the same latent space for each layer $\ell$. Although this design yields excellent test accuracies that are comparable to those of BP-trained models (Wu et al., 2022), it has at least two unconventional and perhaps mysterious characteristics. First, AL involves projecting a one-hot-encoded target variable $y$ into a latent vector $t_1$ and then transforming $t_1$ back into $y$. Interestingly, the length of $t_1$ is sometimes greater than the number of classes. This process corresponds to building an autoencoder whose bottleneck layer is larger than the input and output layers. Although this unconventional approach works surprisingly well in practice (Wu et al., 2022; Kao & Chen, 2021), the fundamental reasons for this are still unclear. Second, when converting a neural network into its AL form, we sometimes need to create extra fully connected layers. In contrast, our design is more natural because we need neither the autoencoder nor the extra fully connected layers.

Finally, since each component has its own local objective, we can parallelize the training procedure by means of pipelining. We use the network illustrated in Figure 3 as an example. Let Task $\ell$ denote the entire forward and backward process in layer $\ell$; then, we can illustrate the pipelining process as shown in Table 1. Specifically, in the first time unit $t_1$, component 1 uses the first batch ($B_1$) to perform Task 1. At $t_2$, component 2 performs Task 2 based on $B_1$, and component 1 continues to performing Task 1 based on the second batch ($B_2$). Starting at $t_3$, all three components can perform forward propagation, backward propagation, and parameter updating simultaneously. However, we have shown here only that parallelization by means of pipelining is feasible; implementation of the pipeline mechanism is left for future work.

## 3 EXPERIMENTS

We compare Delog-SCL with three baselines using different neural networks on different datasets. The baseline models include BP, the Early Exit mechanism introduced in Section 1, and AL, a state-of-the-art method for BP decoupling in terms of the test accuracy. We test three neural networks: a vanilla convolutional neural network (vanilla ConvNet), the VGG network (Simonyan & Zisserman, 2014), and the residual network (ResNet) (He et al., 2016). The experimental datasets include CIFAR-10 (consists of $50,000$ color training images and $10,000$ test images; each image belongs to 1 of 10 classes), CIFAR-100 (consists of $50,000$ color training images and $10,000$ test images; each image belongs to 1 of 100 classes), and Tiny-ImageNet (consists of $100,000$ color training images, $10,000$ validation images, and $10,000$ test images; each image belongs to 1 of 200 classes).

### 3.1 ACCURACY COMPARISON

Table 2: A comparison of the test accuracies (mean $\pm$ standard deviation) of different methodologies when using different neural network architectures on CIFAR-10. We highlight the winner among the non-BP methodologies in bold face. We mark a methodology with a † symbol if the test accuracy of this methodology is higher than that of BP.

|  | Vanilla ConvNet | VGG | ResNet |
|---|---|---|---|
| BP | $86.85 \pm 0.57$ | $93.02 \pm 0.03$ | $93.95 \pm 0.11$ |
| Early Exit | $83.16 \pm 0.33$ | $91.28 \pm 0.15$ | $89.63 \pm 0.34$ |
| AL | $\mathbf{86.98} \pm 0.24$ † | $93.22 \pm 0.12$ † | $91.33 \pm 0.09$ |
| Delog-SCL | $\mathbf{86.98} \pm 0.33$ † | $\mathbf{93.42} \pm 0.11$ † | $\mathbf{92.78} \pm 0.11$ |

Table 3: A comparison of the test accuracies of different methodologies when using different neural network architectures on CIFAR-100. We follow the same notations used in Table 2.

|  | Vanilla ConvNet | VGG | ResNet |
|---|---|---|---|
| BP | $58.68 \pm 0.13$ | $72.58 \pm 0.39$ | $73.59 \pm 0.11$ |
| Early Exit | $50.64 \pm 0.44$ | $71.11 \pm 0.95$ | $64.48 \pm 0.41$ |
| AL | $53.06 \pm 0.15$ | $72.43 \pm 0.27$ | $67.53 \pm 0.32$ |
| Delog-SCL | $\mathbf{59.63} \pm 0.37$ † | $\mathbf{73.14} \pm 0.30$ † | $\mathbf{70.41} \pm 0.27$ |

Table 4: A comparison of the test accuracies of different methodologies when using different neural network architectures on Tiny-ImageNet. We follow the same notations used in Table 2.

|  | VGG | ResNet |
|---|---|---|
| BP | $48.30 \pm 0.14$ | $49.71 \pm 0.18$ |
| Early Exit | $46 \pm 0.18$ | $40 \pm 0.34$ |
| AL | $\mathbf{49.06} \pm 0.14$ † | $44.83 \pm 0.15$ |
| Delog-SCL | $48.95 \pm 0.17$ † | $\mathbf{46.87} \pm 0.26$ |

Table 2 shows the test accuracies of the various methods on the CIFAR-10 dataset. The simple Early Exit mechanism can be used to learn the relationship between an image and its corresponding class. However, the test accuracies of Early Exit are much worse than those of BP. Both AL and our proposed Delog-SCL yield better test accuracies than BP based on the Vanilla ConvNet and VGG architectures. However, when ResNet is used, BP yields the highest test accuracy. If we compare only the methods that involve BP decomposition, Delog-SCL performs the best among them.

We also tested BP, Early Exit, AL, and Delog-SCL on CIFAR-100. The results, as shown in Table 3, are similar to those on CIFAR-10: Delog-SCL performs better than both AL and BP based on Vanilla ConvNet and VGG, whereas Delog-SCL performs worse than BP when ResNet is used. These results are also consistent with those reported in (Kao & Chen, 2021; Wu et al., 2022).

Table 4 gives the results obtained on Tiny-ImageNet. When VGG is used, both AL and Delog-SCL outperform BP. However, for ResNet, BP performs much better.

Delog-SCL is stable in training, as can be shown by Figure 5 in the Appendix.

### 3.1.1 Discussion on accuracy comparison

When BP is used, all parameters are updated to minimize a global objective – the residual between the prediction $\hat{y}$ and the target $y$. On the other hand, methods to decouple end-to-end backpropagation, such as Delog-SCL and AL, are composed of many local objectives, which may differ from the global objective. Therefore, it is surprising that Delog-SCL and AL outperform BP for some network structures. The authors of AL proposed several conjectures to explain this remarkable result. First, projecting the feature vector $x$ and the target $y$ into the same latent space may be helpful. Second, the autoencoder may implicitly perform some feature extraction and regularization. Third, overparameterization may be helpful for optimization (Arora et al., 2018; Chen & Chen, 2020). However, the first and second conjectures only apply to AL but not to Delog-SCL, but Delog-SCL still yields better accuracies than BP and AL in vanilla ConvNet and VGG. Therefore, the above conjectures may not fully explain the success of Delog-SCL. Further investigation will be needed to uncover the fundamental reasons.

As for ResNet, its authors state that the main effect of a residual is not about promoting gradient flows (He et al., 2016). Instead, ResNet performs better than vanilla ConvNet because the latent representations $H_\ell$ and $H_{\ell+1}$ at deep neighboring layers $\ell$ and $\ell + 1$ are likely similar. Regular nonlinear transformations may be difficult to approximate an (almost) identical mapping from $H_\ell$ to $H_{\ell+1}$. However, the residual connection sets $H_{\ell+1}$ to be $f(H_\ell) + H_\ell$. Even if $f()$ is a nonlinear function, a solver is easier to make $H_{\ell+1} \approx H_\ell$ by making $f(H_\ell) \approx 0$. The property that $H_{\ell+1} \approx H_\ell$ is likely true when a network is deep. However, when using Delog-SCL or AL, each local network is short, so Delog-SCL and AL are unlikely to take advantage of the residual connections. As a result, optimizing a ResNet by BP usually gives better results than by BP-decoupling methods, such as Delog-SCL and AL.

### 3.2 Number of effective parameters

Table 5: A comparison of the numbers of parameters used during training and testing with different methodologies (using CIFAR-10 as an example)

|  | VGG | | ResNet | |
|---|---|---|---|---|
|  | Training | Testing | Training | Testing |
| BP | **11.9M** | **11.9M** | **11.2M** | **11.2M** |
| AL | 284.9M | 13.9M | 325.1M | 14.4M |
| Delog-SCL | 70.1M | **11.9M** | 72.4M | **11.2M** |

Table 5 shows the numbers of parameters required during training and inference for VGG and ResNet (using CIFAR-10 as an example).

For BP, the training and testing stages involve the same set of parameters. In contrast, the training process of AL requires additional bridge functions and encoding functions, which are not used

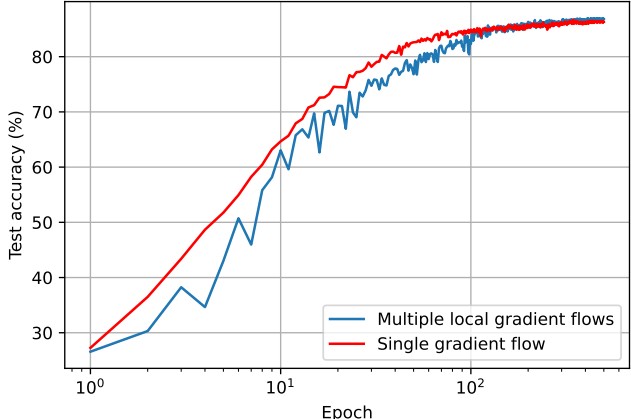

Figure 4: Training Delog-SCL on CIFAR-10 using multiple short gradient flows vs using single long gradient flow.

during testing. So, the number of training parameters is much larger than the number of parameters required for BP. During testing, the extra fully connected layers in AL also require more parameters than are needed in BP. Finally, Delog-SCL and BP have the same number of testing parameters. However, during training, Delog-SCL needs the projection heads $g_\ell$ for each component, so the number of required parameters during training is more than for BP but much fewer than for AL.

The difference of the parameter counts may also be reflected on the training and inference speed. We show the practical training and testing time of different methods in Table 7 in the Appendix.

### 3.3    MULTIPLE SHORT GRADIENT FLOWS ACCELERATE LEARNING

This section shows that dividing a long gradient flow into multiple short ones accelerates learning.

Referring to Figure 3, our proposed Delog-SCL uses a local supervised contrastive loss to create a short local gradient flow $\mathcal{L}_\ell^{SC}$. We compare the standard Delog-SCL with a modification where only a single long gradient flow is used. In the compared baseline, we enable the global objective $\mathcal{L}^{OUT}$ to pass through the entire network and remove all local supervised contrastive losses $\mathcal{L}_\ell^{SC}$.

The results are shown in Figure 4. We label the original Delog-SCL as "multiple short gradient flows" and the modification with single long gradient flow as "single gradient flow". Using multiple short gradient flows accelerates the learning speed, especially in the first 100 epochs.

### 3.4    THE EFFECT OF BATCH SIZE AND PROJECTION HEAD

We also experimented with how the batch size and the type of projection head influence learning. The experimental results show that a larger batch size improves the learning quality, which is consistent with previous studies (Chen et al., 2020; Henaff, 2020; Bachman et al., 2019). As for the projection head, using a nonlinear function benefits the representation quality of layers before it. The result also matches the experiments conducted in Chen et al. (2020).

The experimental details of batch size and projection heads are presented in Section A.3 and Section A.4 in Appendix.

## 4    RELATED WORK

Studies on alternatives to BP mostly aim to address optimization and performance issues, such as gradient vanishing/explosion and training costs. We review some of these works that have particularly focused on the creation of local objectives and local gradient flows.

Table 6: A comparison of the properties of BP, AL, and Delog-SCL

|  | Number of affiliated parameters | Structure flexibility | Length of gradient flows | Allows pipelined training |
|---|---|---|---|---|
| BP | 0 | High | Long | False |
| AL | Many | Medium | Short | True |
| Delog-SCL | Few | Medium | Short | True |

The first type of BP alternative is target propagation (Lee et al., 2015; Meulemans et al., 2020; Manchev & Spratling, 2020; Bengio, 2014), which assigns a local target for each layer via feedback (inverse) mapping. Such a methodology can alleviate the problem of vanishing/exploding gradients since each gradient flow is short. However, the parameters are still updated in a layerwise fashion, so it could be challenging to learn the parameters in different layers simultaneously.

Methods of the second type model BP as a constrained optimization problem, in which the output of one layer is forced to equal the input to the next layer (Gotmare et al., 2018; Marra et al., 2020). Such a design shortens the gradient flows and enables parallel parameter updates. However, the experimental results show that the test accuracies are lower than that of standard BP.

Methods of the third type determine the local objectives through transformations of the target. A representative method of this type is AL Wu et al. (2022); Kao & Chen (2021), which transforms both the feature vector $x$ and the target $y$ into the same set of latent spaces. To the best of our knowledge, AL is the only existing method that can achieve BP decoupling for a wide range of network architectures and yield test accuracies that are comparable to those obtained with BP.

Our proposed Delog-SCL is motivated by both AL and Greedy InfoMax (GIM) (Löwe et al., 2019), which uses the contrastive loss as each local objective. However, GIM targets self-supervised learning tasks, whereas our Delog-SCL can handle supervised learning tasks because Delog-SCL uses the supervised contrastive loss in the hidden layers and the distance between the predicted and observed targets in the output layer.

Since only Delog-SCL and AL yield test accuracies comparable to those of BP, in Table 6, we further compare the properties of these three methods. First, BP requires no affiliated parameters because all parameters collaborate to reduce the global loss. BP can be applied to almost all kinds of neural networks. However, its gradient flow is long (especially when the network is deep), and it is challenging to achieve pipelined training with this method. AL requires transforming both the feature vector $x$ and the target $y$ alongside each other. As a result, AL usually requires additional fully connected layers, resulting in a large number of affiliated parameters and less structural flexibility. However, each gradient flow in AL is short, and parameters in different layers can be updated simultaneously via pipelined training. Finally, because Delog-SCL requires computing the supervised contrastive loss in each hidden layer, this method also needs additional affiliated parameters (although much fewer than AL) during training. The introduction of the supervised contrastive loss also adds complexity in the network design. The advantages of Delog-SCL are similar to those of AL: the gradient flows are short, and parallel parameter updating is possible (via pipelining).

## 5 CONCLUSION

This paper presents Delog-SCL, a new methodology for decoupling the components of the BP process in a neural network. Delog-SCL may address various optimization issues (e.g., vanishing/exploding gradients and unstable gradients in the early layers) resulting from the long gradient flows in deep neural networks. We report experiments conducted to show that Delog-SCL's predictive power is comparable to (and frequently better than) that of either BP or AL, which is a state-of-the-art alternative to BP. Delog-SCL is more flexible than AL because Delog-SCL does not require additional fully connected layers, whereas AL usually does. Therefore, Delog-SCL is a natural substitute for AL and could be a promising alternative to BP.

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

## A  Appendix

### A.1  Test accuracy vs epoch

Figure 5 compares Delog-SCL and BP on vanilla CNN in terms of their dynamics of test accuracy when the epoch increases. First, the test accuracy of Delog-SCL improves stably. Second, the Delog-SCL outperforms BP after approximately 100 epochs. BP is better than Delog-SCL at the beginning, likely because all the parameters in BP are updated to minimize a global objective. On the contrary, most of the parameters in Delog-SCL are updated to fit local objective functions, which usually have no direct access to the target variable.

Different hyperparameter settings may lead to slightly different curves. However, most of them follow a similar pattern. Experiments on other datasets (CIFAR-10 and tiny-ImageNet) for the VGG network structure also show similar trends.

### A.2  A comparison of training and test time of BP, AL, and Delog-SCL

Table 7 compares Delog-SCL, BP, and AL in terms of their practical training and testing seconds per epoch on ConvNet, VGG, and ResNet using the CIFAR-10 dataset. The empirical training and testing speed of Delog-SCL and BP are extremely close. However, AL is apparently slower than the other two.

The experiments are tested on the Container Compute Service with NVIDIA Tesla V100 GPU.

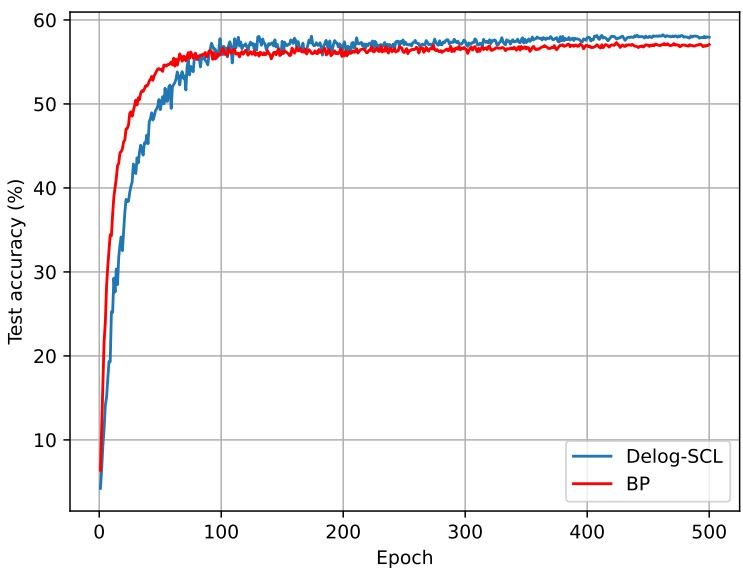

Figure 5: Epoch vs test accuracy for BP and Delog-SCL on CIFAR-100 using vanilla ConvNet.

Table 7: A comparison of the training and testing seconds (mean $\pm$ standard deviation) per epoch with different methodologies (using CIFAR-10 as an example)

|  | ConvNet | | VGG | | ResNet | |
|---|---|---|---|---|---|---|
|  | Training | Testing | Training | Testing | Training | Testing |
| Delog-SCL | $16.62 \pm 0.71$ | $\mathbf{1.58 \pm 0.01}$ | $51.63 \pm 0.09$ | $\mathbf{2.67 \pm 0.01}$ | $42.14 \pm 0.54$ | $\mathbf{2.34 \pm 0.01}$ |
| BP | $\mathbf{15.74 \pm 0.33}$ | $1.59 \pm 0.02$ | $\mathbf{50.05 \pm 0.06}$ | $\mathbf{2.67 \pm 0.01}$ | $\mathbf{42.05 \pm 0.55}$ | $2.35 \pm 0.02$ |
| AL | $22.81 \pm 0.56$ | $1.82 \pm 0.02$ | $71.40 \pm 0.81$ | $3.19 \pm 0.02$ | $56.62 \pm 1.29$ | $2.39 \pm 0.01$ |

### A.3 A LARGER BATCH SIZE IMPROVES THE LEARNING QUALITY

We also tested how the batch size influences the test accuracy. As shown in Table 8, performing Delog-SCL training using a large batch size is helpful, and the improvement on VGG is more evident than in other networks. This finding is consistent with the results reported in previous studies, e.g., Chen et al. (2020); Henaff (2020); Bachman et al. (2019), in which the authors noted that because a larger batch tends to include more negative pairs (as shown in Equation 2), the model has access to more information that can be used to distinguish positive pairs from negative pairs. Although other studies, e.g., Mitrovic et al. (2020), have shown that the number of negative pairs may not be critical to the improvement of the test accuracy, most studies tend to agree that a larger batch size leads to better results.

Table 8: The test accuracies of Delog-SCL when using different batch sizes on CIFAR-10.

| Batch Size | VGG | ResNet |
|---|---|---|
| 32 | $92.67 \pm 0.10$ | $92.54 \pm 0.14$ |
| 128 | $93.11 \pm 0.16$ | $92.53 \pm 0.11$ |
| 1024 | $93.42 \pm 0.11$ | $92.78 \pm 0.11$ |

Table 9: The test accuracies when using different projection heads on CIFAR-10.

| Type of projection head | VGG | ResNet |
|---|---|---|
| Identity | 79.2 | 84.2 |
| Linear | 90.4 | 90.2 |
| MLP | 93.0 | 92.4 |

### A.4 A NONLINEAR PROJECTION HEAD BENEFITS THE REPRESENTATION QUALITY OF THE LAYER BEFORE IT

This section presents the influence of different projection heads. Table 9 compares the accuracies of VGG and ResNet on CIFAR-10 when 3 different types of projection heads are used: identity mapping (i.e., $g_\ell\left(\boldsymbol{r}_\ell^{(i)}\right) = \boldsymbol{r}_\ell^{(i)}$), linear mapping (i.e., $g_\ell\left(\boldsymbol{r}_\ell^{(i)}\right) = \boldsymbol{w}^T\boldsymbol{r}_\ell^{(i)} + w_0$), and the default mapping based on a multilayer perceptron (MLP). The results are similar to those reported in (Chen et al., 2020): the MLP mapping shows a $2.6\%$ improvement over linear projection, which outperforms identity projection by over $10\%$.

Using an MLP as the projection head is beneficial likely because the information loss induced by the contrastive loss is more severe when a simple projection head is used (Chen et al., 2020). In particular, since a projection head $g_\ell$ (refer to Figure 3 and Figure 1) maximizes the agreement between augmented images, $g_\ell$ may remove information relevant to image rotation, flipping, and other data augmentation operations, which could be useful for downstream tasks. When a simple projection head $g_\ell$ is used, the information contained in $\boldsymbol{r}_\ell^{(i)}$ will be similar to that in $\boldsymbol{z}_\ell^{(i)}$, which means that $\boldsymbol{r}_\ell^{(i)}$ is invariant to data augmentation. On the other hand, when a complex projection head such as an MLP is used, the information in $\boldsymbol{r}_\ell^{(i)}$ may be very different from that in $\boldsymbol{z}_\ell^{(i)}$. As a result, even if $\boldsymbol{z}_\ell^{(i)}$ loses information relevant to data augmentation, $\boldsymbol{r}_\ell^{(i)}$ may still preserve this information.

### A.5 PSEUDO CODE

Here we provide a PyTorch pseudocode for the creation of the local supervised contrastive losses (Algorithm 1) and Delog-SCL (Algorithm 2).

**Algorithm 1:** PyTorch-like pseudocode for $L^{sc}$

```python
import torch
import torch.nn as nn
class SupConLoss(nn.Module):
  def __init__(self, dim):
    super.__init__()
    self.linear =nn.Sequential(nn.Linear(dim, 512), nn.ReLU(),
      nn.Linear(512, 1024))
    self.temperature =0.1
  def forward(self, x, label):
    x = self.linear(x)
    x = nn.functional.normalize(x)
    label =label.view(-1, 1)
    bsz = label.shape[0]
    mask =torch.eq(label, label.T).float()
    anchor_mask =torch.scatter(torch.ones_like(mask), 1, torch
      .arange(bsz).view(-1, 1), 0)
    logits =torch.div(torch.mm(x, x.T), self.temperature)
    deno =torch.exp(logits)*anchor_mask
    prob =logits -torch.log(deno.sum(1, keepdim=True))
    loss =- (anchor\_mask *mask *prob).sum(1)/mask.sum()
    return loss.view(1, bsz).mean()
```

**Algorithm 2:** PyTorch-like pseudocode for Delog-SCL

```python
import torch
import torch.nn as nn
# A 3-layer vanilla ConvNet example for Delog-SCL
class CNN_DelogSCL(nn.Module):
  def __init__(self, dim):
    super.__init__()
    CNNs = [ ]
    losses =[]
    channels =[3, 128, 256, 512]
    self.shape =32
    for i in range(3):
      CNNs.append(nn.Sequential(nn.Conv2d(channels[i],
        channels[i+1], padding=1), nn.ReLU()))
      losses.append(SupConLoss(self.shape*self.shape*channels[
        i+1]))
    self.CNN =nn.ModuleList(CNNs)
    self.loss =nn.ModuleList(losses)
    self.fc =nn.Sequential(flatten(), nn.Linear(self.shape*
      self.shape*channels[-1], 10))
    self.ce =nn.CrossEntropyLoss()
  def forward(self, x, label):
    loss =0
    for i in range(3):
      # .detach() prevents the computation graph from
       propagating gradients to the next layer
      x = self.CNN[i](x.detach())
      if self.training:
        loss +=self.loss[i](x, label)
    y = self.fc(x.detach())
    if self.training:
      loss +=self.ce(y, label)
      return loss
    return y
```

