# OpenReview forum: "Cutting Long Gradient Flows: Decoupling End-to-End Backpropagation Based on Supervised Contrastive Learning"
_ICLR.cc/2023/Conference — Submitted to ICLR 2023_

### Official Review · Reviewer_UdqE · 2022-10-24

**Confidence:** 4
**Correctness:** 4
**Technical Novelty And Significance:** 2
**Empirical Novelty And Significance:** 3
**Recommendation:** 5

**Clarity, Quality, Novelty And Reproducibility:**

**Clarity and quality.** While the methodology is clear, the writing would benefit from a bit of polishing. Parts like the abstract, or short phrases extremely similar to the table captions (e.g. Table 5 and below) could be improved.

**Novelty.** Here lies my biggest concern: the proposed method is extremely similar (to my eyes) to Greedy InfoMax. If I understood correctly, the only difference between both approaches is that the proposed method uses _supervised_ contrastive learning, while Greedy InfoMax uses _unsupervised_ contrastive learning. The authors should better clarify what makes this work novel.

**Reproducibility.** There is no code provided until paper acceptance, neither there is an appendix. Therefore, the paper lacks all the experiemtal details to perfectly reproduce the reported results.

**Strength And Weaknesses:**

**Strengths:**
- S1. The approach is clear and simple, and provides results at times better than back-propagation, which I find remarkable.
- S2. The proposed method is a clear improvement over the competing method, beating it in every aspect as far as I can tell.
- S3. The ablation study is nice and agrees with existing knowledge in contrastive learning.

**Weaknesses:**
- W1. The motivation is rather weak. While the asynchronous training is interesting, having long gradients does not seem as much of a problem, as it can be overcome with architectural changes (which is clearly shown by the performance difference of the existing approaches in the ResNet experiments).
- W2. Despite contrastive learning being quite intense in computations, there are no mentions to memory and time consumption.
- W3. Needed work is left as future work. Specifically, pipelining as well as understanding where comes the improvement in performance come from.
- W4. Despite talking all the paper about gradients, there is not a single experiment exploring the gradients and showing that there was actually a problem in the chosen experimental setups.

**Comments:**
- C1. While parallelism is trivialized in the proposed approach, pipelining is also possible in traditional methods, as demonstrated by companies such as [OpenAI](https://openai.com/blog/techniques-for-training-large-neural-networks/).
- C2. Doesn't the fact that Early Exit perform well in CIFAR-100 show that it is not an appropriate experimental setup?

**Summary Of The Paper:**

As neural networks get deeper, end-to-end backpropagation requires computing the gradients along longer paths, thus making gradient computations more prone to numerical instabilities and vanishing/exploding gradients. This paper proposes a new approach which divides the network into different submodules that are independently trained. Specifically, inner modules are trained using supervised contrastive learning, while the last layer is trained as usual. This enables the computation of short gradients, as well as the asynchronous training of modules, which beats the current state-of-the-art method and obtains comparatively results with end-to-end backpropagation.

**Summary Of The Review:**

While the proposed approach is novel in the supervised setting, I have serious concerns of its novelty with respect to Greedy InfoMax. While this would not directly imply a rejection, the deferred extra work and the lack of proper experiments to fully understand the benefits and hurdles of the proposed approach in the specific setting it is used for, makes me lean towards rejection.

---

> ### Author Response · Authors · 2022-11-24
> **Thank you for your insightful review and advices!**
>
> **Q1. The relationship between Greedy InfoMax and Delog-SCL.**
>
> Greedy InfoMax leverages contrastive learning (CL), and Delog-SCL uses supervised contrastive learning (SCL). The difference is huge in practice when applying them. CL involves no instance labels, so creating positive and negative pairs relies on data augmentation. Unfortunately, data augmentation is straightforward only on limited tasks (mostly in computer vision). Therefore, Greedy InfoMax can only be applied to these limited tasks. On the other hand, SCL uses instance labels to generate positive and negative pairs. Therefore, Delog-SCL can be used on much broader types of tasks.
>
> **Q2. Having long gradients does not seem as much of a problem, as it can be overcome with architectural changes.**
>
> Indeed, architectural changes, e.g., residual connections and non-saturated activation functions, partially alleviate optimization difficulties. However, they do not fully solve these issues, so various experiments and models were still proposed after the invention of ResNet in 2016 (e.g., [r1, r2, r3]).
>
> We address these optimization issues from a different perspective – decoupling a long gradient flow into multiple short ones. As for why ResNet seems to work well when trained via BP but less optimal when trained by Delog-SCL or AL, it is likely because residual connections excel at approximating a (near-)identity mapping. As explained by the authors of ResNet (Sec. 4.1 of He et al., 2016), ResNet performs better than a vanilla ConvNet likely because the latent representations $H_{\ell}$ and $H_{\ell+2}$ for two deep layers $\ell$ and $\ell+2$ are likely similar. With residual connection, finding $w_{\ell}$ and $w_{\ell+1}$ (parameters for layer $\ell$ and layer $\ell+1$) such that $H_{\ell+2}=H_{\ell}+a(w_{\ell+1}^T a(w_{\ell}^T H_{\ell} ))\approx H_{\ell}$ is easy ($a$ is an activation function) -- simply let $a(w_{\ell+1}^T a(w_{\ell}^T H_{\ell} )) \approx 0$. However, if without residual connections, finding $w_{\ell}$ and $w_{\ell+1}$ such that $H_{\ell+2}=a(w_{\ell+1}^T a(w_{\ell}^T H_{\ell} )) \approx H_{\ell}$ might be difficult. The property that $H_{\ell+2} \approx H_{\ell}$ is more likely to be true when a network is deep. However, when using Delog-SCL or AL, each local network is short, so Delog-SCL and AL are unlikely to take advantage of the residual connections. As a result, optimizing a ResNet by BP usually gives better results than by the BP-decoupling methods, such as Delog-SCL and AL.
>
> [r1] Activated Gradients for Deep Neural Networks. IEEE Transactions on Neural Networks and Learning Systems, 2021.
>
> [r2] Which neural net architectures give rise to exploding and vanishing gradients? NeurIPS 2018.
>
> [r3] Hexpo: A vanishing-proof activation function. IJCNN 2017.
>
> **Q3. Memory and time consumption should be reported for supervised contrastive learning.**
>
> The training time of Delog-SCL is close to that of BP. We added these results in Table 7 in the Appendix.
>
> The memory usage of Delog-SCL during training is indeed larger, as demonstrated in Table 5. However, during testing, Delog-SCL and BP have identical network structures (so the memory usage is the same).
>
> **Q4. Experiments on exploring the gradients should be added to show that long gradient flows are indeed an issue.**
>
> We added an experiment in Section 3.3 to show that dividing a long gradient flow into multiple short ones accelerates learning.
>
> Our proposed Delog-SCL uses a local supervised contrastive loss to create a short local gradient flow $L_{\ell}^{SC}$ (referring to Figure 3). We compare the standard Delog-SCL with a modification where only a single long gradient flow is used. Specifically, we enable the global objective $L^{OUT}$ to pass through the entire network and remove all local supervised contrastive losses $L_{\ell}^{SC}$.
>
> The results are shown in Figure 4. We label the original Delog-SCL as "multiple short gradient flows" and the modification with single long gradient flow as "single gradient flow". Using multiple short gradient flows accelerates the learning speed, especially in the first 100 epochs.
>
> **Q5. Pipelining is also possible in traditional methods, as demonstrated by OpenAI.**
>
> The pipelining demonstrated by OpenAI is different from (but not contradicting) ours. Although the Pipeline Parallelism (e.g., GPipe) mentioned by OpenAI looks similar to Delog-SCL because both methods attempt to simultaneously conduct operations of different layers, Pipeline Parallelism still follows standard backpropagation to update parameters, so the bubbles in the pipeline are unavoidable. On the contrary, the pipeline of Delog-SCL contains no bubbles, so training is more efficient than GPipe.
>
> **Q6. Experiments of Early Exit on different datasets should be included.**
>
> Added. Please see Table 2, Table 3, and Table 4.
>
> **Q7. The source code is not provided for reproducibility.**
>
> We uploaded the source code. In addition, we included pseudo-code in Appendix A.5.

---

> > ### Author Response · Authors · 2022-11-24
> > **Looking forward to hearing from you!**
> >
> > Dear reviewer UdqE,
> >
> > We want to send a friendly reminder for the discussion. Here is a summary of our response for your valuable feedback!
> >
> > * We compared Delog-SCL with Greedy InfoMax.
> > * We explained why long gradients may still be an issue even with architectural changes.
> > * We reported memory consumption (Table 5) and training time (Table 7) for supervised contrastive learning.
> > * We added experiments on exploring the gradients (Sec 3.3; Fig 4).
> > * We explained the difference between the pipelining demonstrated by OpenAI and ours.
> > * We added results on Early Exit (Table 2, Table 3, Table 4).
> > * We uploaded experimental code and provided pseudocode in A.5
> >
> > We thank you again for the valuable comments, and we would appreciate it if you could reconsider the evaluation of our work based on our response. We are also happy to extend our response if you have any other concerns.
> >
> > Thanks.

---

> > > ### Author Response · Authors · 2022-11-30
> > > **Looking forward to hearing from you!**
> > >
> > > Dear reviewer UdqE,
> > >
> > > We would very much appreciate getting a response from you.
> > >
> > > Thanks.

---

> > > > ### Comment · Reviewer_UdqE · 2022-11-30
> > > > **Response to the authors**
> > > >
> > > > Dear Authors,
> > > > I am really sorry for not responding to you. I wrote down to the reviewing team, but forgot to reply to you.
> > > >
> > > > I appreciate very much all the changes you have made to the manuscript. All my doubts have been solved, and I think it is remarkable that you obtained these good results with an alternative way of training a neural network.
> > > >
> > > > However, my two main concerns still remain. First, the novelty of the method is rather limited when compared with Greedy InfoMax, as their differences are mainly those differences between unsupervised and supervised contrastive learning. Second, the use-case/focus of the paper is a bit weak, as widely-adopted techniques to cope with exploding/vanishing gradients (such as residual connections) are as effective as the proposed method.
> > > >
> > > > I will update my score to weak reject.

---

> > > > > ### Author Response · Authors · 2022-12-01
> > > > > **The applicable scenario of Greedy InfoMax is limited; ReLU and residuals do not fully address the issue of vanishing gradient**
> > > > >
> > > > > Dear reviewer UdqE,
> > > > >
> > > > > Thanks for the reply.
> > > > >
> > > > > Regarding the comparison between Greedy InfoMax and Delog-SCL, Greedy InfoMax can only be used on limited applications because Greedy InfoMax requires data augmentation to generate positive instance pairs. Unfortunately, data augmentation is primarily applicable in computer vision applications. There is no principal way to augment data instances in other domains. In contrast, data augmentation is an optional process for Delog-SCL. So, Delog-SCL can be applied to much broader types of applications.
> > > > >
> > > > > As for a common belief that residual connections and ReLU address vanishing/exploding gradients, they do not fully address these issues. The main effect of residual connections is not about promoting gradient flows (Sec. 4.1 in He et al.); please refer to the reply of Q2 for details. ReLU still suffers from the vanishing gradient because the derivative of ReLU is zero when the input argument is negative. All in all, no technique fundamentally solves the issue of vanishing/exploding gradients; they only mitigate the issue. Our work tackles the problem from a different perspective -- shorten the gradient flows. Therefore, the new design deserves to be discussed in ICLR.
> > > > >
> > > > > Thanks.

---

### Official Review · Reviewer_Xd2U · 2022-10-24

**Confidence:** 4
**Correctness:** 3
**Technical Novelty And Significance:** 3
**Empirical Novelty And Significance:** 3
**Recommendation:** 6

**Clarity, Quality, Novelty And Reproducibility:**

- The main body of the paper is very clear, and of high quality.

- The paper is not reproducible, as no code was included, and hyper parameters such as learning rates are not included in the paper or abstract. I recommend that the authors release the code, and additionally describe experiments in detail in an appendix.

**Strength And Weaknesses:**

# Strengths:

- The paper addresses a question that remains relevant: how do we best minimize losses in deep neural networks? How do we prevent gradients from vanishing?

- The paper is very clear and well-written, and easy to read.

# Weaknesses:

- The authors did not include the code with their submission, and mentioned no intent to do so. There is also no appendix with experimental details such as layer sizes and learning rates. Reproducing the experiments would be practically impossible.

- Besides the methods described in the submission, skip connections (e.g. in highway networks https://arxiv.org/pdf/1505.00387.pdf or residual networks) are an effective tool to reduce vanishing gradient problems. The authors should highlight this in their literature review, and ideally also perform experiments to check if adding skip connections would remove the need for the method described in this paper. It is interesting that the authors find that their method (as well as the AL benchmark method) do not outperform backpropagation when ResNets are used. This suggests that skip connections may remove the need for methods such as the one proposed in this paper, as well as the AL baseline. It would be an interesting experiment to check how e.g. VGG with added skip connections performs with and without the proposed Delog-SCL method. Skip connections have also been used in LSTMs, e.g. in https://aclanthology.org/C16-1020.pdf and https://ojs.aaai.org/index.php/AAAI/article/view/4613/4491 .

- The authors write “We omitted Early Exit because this method performs much worse on more complicated datasets”. In my opinion, it would be better to include the Early Exist performance numbers, and not need the explanation of why Early Exit was not included.



**Summary Of The Paper:**

UPDATE: FOLLOWING THE AUTHOR'S RESPONSE, I CHANGE MY SCORE FROM 5 to 6.

The authors describe a method to reduce the problems associated with gradients in settings where backpropagation happens through many layers. In the context of supervised contrastive learning, the authors propose to compute contrastive losses not only at the output layer, but also at intermediate layers. This allows the authors to segment the neural network into blocks that each have their own losses, and can therefore be trained nearly independently (of course, layers closer to the output still must adapt to changes in the closer-to-input layers during training). The authors compare their method with the Associated Learning method, and find that their method performs about as well, but is easier to work with. The authors also compare with vanilla back propagation, and find that their method (as well as the associated learning baseline) outperform back propagation on a number of tasks and network layer types (but not on residual networks).

**Summary Of The Review:**

UPDATE: FOLLOWING THE AUTHOR'S RESPONSE, I CHANGE MY SCORE FROM 5 to 6.

For now, I mark this submission "5: marginally below the acceptance threshold".

The main idea presented in the paper is interesting and useful. However, before the paper can be accepted, the authors should improve the reproducibility of the paper by releasing code and describing experiments in more detail in an appendix. The authors should also investigate whether skip connections could achieve the same that their method does.

---

> ### Author Response · Authors · 2022-11-18
> **Thank you for your insightful review and advices!**
>
> **Q1: The source code is not provided for reproducibility; hyper-parameters, such as learning rates, are not included.**
>
> We uploaded the source code.  In addition, we included pseudo-code in Appendix A.5.
>
> **Q2: Discussions on skip connections (e.g., highway network or residual network) and their relationship to Delog-SCL and AL should be added.**
>
> The ResNet authors state that a residual's primary effect is not promoting gradient flows (Sec. 4.1 in He et al., 2016). ResNet performs better than a vanilla ConvNet likely because the latent representations $H_{\ell}$ and $H_{\ell+2}$ for two deep layers $\ell$ and $\ell+2$ are likely similar. With residual connection, finding $w_{\ell}$ and $w_{\ell+1}$ (parameters for layer $\ell$ and layer $\ell+1$) such that $H_{\ell+2}=H_{\ell}+a(w_{\ell+1}^T a(w_{\ell}^T H_{\ell} ))\approx H_{\ell}$ is easy ($a$ is an activation function) -- simply let $a(w_{\ell+1}^T a(w_{\ell}^T H_{\ell} )) \approx 0$. However, if without residual connections, finding $w_{\ell}$ and $w_{\ell+1}$ such that $H_{\ell+2}=a(w_{\ell+1}^T a(w_{\ell}^T H_{\ell} )) \approx H_{\ell}$ might be difficult. The property that $H_{\ell+2} \approx H_{\ell}$ is more likely to be true when a network is deep. However, when using Delog-SCL or AL, each local network is short, so Delog-SCL and AL are unlikely to take advantage of the residual connections. As a result, optimizing a ResNet by BP usually gives better results than by the BP-decoupling methods, such as Delog-SCL and AL.
>
> **Q3: Experimental results on "Early Exit" should be included.**
>
> Added to Table 2, Table 3, and Table 4.

---

> > ### Author Response · Authors · 2022-11-24
> > **Looking forward to hearing from you**
> >
> > Dear reviewer Xd2U,
> >
> > We want to send a friendly reminder for the discussion. Here is a summary of our response for your valuable feedback!
> >
> > * We uploaded the source code and provided the pseudocode in Appendix.
> > * We discussed why Delog-SCL/AL could not take advantage of skip connections (Sec 3.1.1).
> > * We included the experimental results on ``Early Exit'' (Table 2, 3, 4).
> >
> > We thank you again for the valuable comments, and we would appreciate it if you could reconsider the evaluation of our work based on our response. We are also happy to extend our response if you have any other concerns.
> >
> > Thanks.

---

> > > ### Comment · Reviewer_Xd2U · 2022-11-29
> > > **Thank you for the response and improvements**
> > >
> > > Dear authors, based on your response, I change my review score from "5" to "6". The paper is now solid enough to be published. In my opinion, it is still not as novel as papers receiving "8" scores.
> > >
> > > I believe that the paper may not make it into ICLR, but would very likely be accepted by a workshop or conference on contrastive learning.

---

> > > > ### Author Response · Authors · 2022-11-30
> > > > **Correction on Q2**
> > > >
> > > > Dear Reviewer Xd2U,
> > > >
> > > > We found a mistake in our earlier response to Q2.  Please see the corrected comments above.
> > > >
> > > > Thanks.

---

### Official Review · Reviewer_XPBU · 2022-10-25

**Confidence:** 4
**Correctness:** 2
**Technical Novelty And Significance:** 2
**Empirical Novelty And Significance:** 2
**Recommendation:** 3

**Clarity, Quality, Novelty And Reproducibility:**

Clarity
------------

The paper lacks clarity on its contribution. This makes it slightly tricky to follow the manuscript.

Quality
-------------
I find the quality of the paper to be good but with lots of room for improvement.

Novelty
------------
I find the work to be novel.

Reproducibility
-------------------
It appears that there has been no code that has been released yet but practically the method is simple and can probably be reproduced.

**Details Of Ethics Concerns:**

Nothing to add.

**Strength And Weaknesses:**

Strengths
-----------

Delog-SCL provides competitive results against Backpropagation and is parameter efficient in comparison to related work (AL). Naturally, as a consequence of gradient decoupling, Delog-SCL circumvents the gradient update locking problem found in Backpropagation.

The algorithm is promising, lots of insights can be teased from this method and it is a fair addition to the literature on decoupled gradient optimization in artificial neural networks.

Weaknesses
----------------
There is a lack of clarity on the novel contributions of the proposed method, Delog-SCL. There is no clear consensus on the prevailing issue that Delog-SCL aims to tackle, a host of issues have been presented, some of which some are unrelated, in particular the Vanishing/Exploding gradient problem. The issue of vanishing and exploding gradients is prematurely explored, for example there is little mention of Delog-SCL being extended to recurrent nets.

Some of the issues highlighted are naturally tackled by contemporary backprop decoupling methods, there was some justification that Delog-SCL is more flexible and simpler than AL, but this was not raised in light of a critical issue (beyond efficient parallelization).

Similarly, the claim that Delog-SCL tackles the issue of unstable gradients in the early layers was mentioned in the introduction and conclusion with little to no explanation on the reasons behind the instability and how this relates to Delog-SCL. This was not explored in the experiments nor methodology section.

**Summary Of The Paper:**

The authors present a novel method that decouples back propagation gradient computation in artificial neural networks. The authors identify issues that result from long gradient flows in back propagation, and propose a method that tackles some issues, which include vanishing/exploding gradients, and update locking. In short, the proposed method cuts long gradient flows into several local gradients with a ‘read-out’ at each layer and a corresponding contrastive loss.

**Summary Of The Review:**

The work is an interesting alternative to the backpropagation algorithm as well as other existing decoupled gradient computation algorithms. However, the biggest shortfall of this manuscript is the lack of clarity on its contributions. The manuscript lacks a problem statement.

---

> ### Author Response · Authors · 2022-11-18
> **Thank you for your insightful review and advices!**
>
> **Q1: More clarification is needed on why Delog-SCL tackles vanishing/exploding gradients and instability gradients in the early layers.**
>
> Vanishing/exploding gradients and unstable gradients in the early layers are primarily caused by the long gradient flows [r1].  Delog-SCL cuts a long gradient flow into multiple smaller gradient flows to address these issues.
>
> Dividing a long gradient flow is challenging because the local objectives may not align with the global objective.  As a result, previous works that attempt to cut long gradient flows usually tested their methods on simple networks with unsatisfactory test accuracies even on simple datasets (e.g., Jaderberg et al., 2017; Czarnecki et al., 2017; Löwe et al., 2019; Wu et al., 2022; Kao & Chen 2021).  Our model is one of the few that yields comparable test accuracies with the models trained by end-to-end backpropagation.
>
> We modify the introduction section to clarify the contribution.
>
> [r1] Hochreiter, Sepp. "The vanishing gradient problem during learning recurrent neural nets and problem solutions." International Journal of Uncertainty, Fuzziness and Knowledge-Based Systems 6.02 (1998): 107-116.
>
> **Q2: Explorations on the gradient-related issues should be added.**
>
> We added an experiment in Section 3.3 to show that dividing a long gradient flow into multiple short ones accelerates learning, which is likely the result of better gradient passing.
>
> Our proposed Delog-SCL uses a local supervised contrastive loss to create a short local gradient flow $L_{\ell}^{SC}$ (referring to Figure 3). We compare the standard Delog-SCL with a modification where only a single long gradient flow is used. Specifically, we enable the global objective $\mathcal{L}^{OUT}$ to pass through the entire network and remove all local supervised contrastive losses $\mathcal{L}_{\ell}^{SC}$
> .
>
> The results are shown in Figure 4. We label the original Delog-SCL as "multiple short gradient flows" and the modification with single long gradient flow as "single gradient flow". Using multiple short gradient flows accelerates the learning speed, especially in the first 100 epochs.
>
> **Q3: Some of the issues highlighted are naturally tackled by contemporary backprop decoupling methods, e.g., AL.**
>
> Indeed, if a methodology can decompose a long gradient flow into multiple small pieces, most of the issues discussed in the introduction are naturally tackled.  However, it is highly challenging to decouple an end-to-end backpropagation without sacrificing test accuracies on the target task.  Most previous works on this line fail to yield comparable test accuracies even on simple neural networks and simple datasets (Jaderberg et al., 2017; Czarnecki et al., 2017; Löwe et al., 2019; Wu et al., 2022; Kao & Chen 2021).  AL is probably the only work that successfully cuts a long gradient flow into small pieces and maintains comparable test accuracies on modern neural networks, e.g., VGG.  However, AL requires a complicated modification of the original network into the "AL form", which includes multiple extra (and likely redundant) fully connected layers that vastly increase the number of parameters.  In contrast, Delog-SCL only moderately reassembles the training network and maintains an identical network structure to BP during testing.
>
> Since decoupling backpropagation may solve the various optimization issues caused by standard BP, but methods to decouple backpropagation either yield worse test accuracies (e.g., Jaderberg et al., 2017; Czarnecki et al., 2017; Löwe et al., 2019; Wu et al., 2022; Kao & Chen 2021) or require complicated modification to the network structure (e.g., Wu et al., 2022), Delog-SCL could be a favorable decoupling method among the alternatives.
>
> **Q4: The source code is not provided for reproducibility.**
>
> We uploaded the source code.  In addition, we included pseudo-code in Appendix A.5.

---

> > ### Author Response · Authors · 2022-11-22
> > **Looking forward to hearing from you**
> >
> > Dear reviewer XPBU,
> >
> > We want to send a friendly reminder for the discussion. Here is a summary of our response for your valuable feedback!
> >
> > * We explained how Delog-SCL tackles vanishing/exploding gradients and instability gradients in early layers.
> > * We conducted experiments to show that multiple short gradient flows accelerate the learning process.
> > * We explained the superiority of Delog-SCL when compared with other backpropagation decoupling methods.
> > * We uploaded the source code and provided the pseudocode in Appendix.
> >
> > We thank you again for your valuable comments, and we would appreciate it if you could reconsider the evaluation of our work based on our response. We are also happy to extend our response if you have any other concerns.
> >
> > Thanks.

---

> > > ### Comment · Reviewer_XPBU · 2022-11-28
> > > **Lack of Clear Contribution in Delog-SCL**
> > >
> > > Thank you for responding to my review. Thank you for also providing the source code and the pseudocode in the Appendix!
> > >
> > > I do acknowledge that Delog-SCL is an interesting alternative to Backpropagation not only from an empirical standpoint but also with regards to issues like exploding/vanishing gradients, gradient instability in early layers, and gradient update locking. However, I find that the paper is relatively shallow in its analysis of these issues and doesn't fully explore these issues in light of Delog-SCL. Each respective issue has been very well studied in literature, particularly exploding/vanishing gradients, and I could argue that in practice, exploding/vanishing gradients are becoming less of an issue in contemporary artificial neural networks, due to ReLU networks.
> > >
> > > I can appreciate that Delog-SCL is a simpler and empirically competitive alternative to AL, which I find to be the key contribution of this work but it is unfortunately not well communicated in this manuscript.
> > >
> > > I find this work to not be tackling a clear problem and performing corresponding analysis on this problem. The lack of clear contribution still stands despite the clarification provided by authors.

---

> > > > ### Author Response · Authors · 2022-11-30
> > > > **Reasons why ReLU (and other famous techniques) does not solve vanishing gradient**
> > > >
> > > > Thanks for the response!
> > > >
> > > > ReLU addresses part of the vanishing/exploding gradients. However, when the input argument $z$ of ReLU is negative, the derivative is zero, which means the vanishing gradient is still an issue. In fact, after Hinton introduced ReLU in 2012, there are still many studies on promoting gradient flows.
> > > >
> > > > ResNet is another methodology commonly described as a solver for vanishing gradients. However, the ResNet authors state that a residual's primary effect is not promoting gradient flows (Sec. 4.1 in He et al., 2016). ResNet performs better than a vanilla ConvNet likely because the latent representations $H_{\ell}$ and $H_{\ell+2}$ for two deep layers $\ell$ and $\ell+2$ are likely similar. With residual connection, finding $w_{\ell}$ and $w_{\ell+1}$ (parameters for layer $\ell$ and layer $\ell+1$) such that $H_{\ell+2}=H_{\ell}+a(w_{\ell+1}^T a(w_{\ell}^T H_{\ell} ))\approx H_{\ell}$ is easy ($a$ is an activation function) -- simply let $a(w_{\ell+1}^T a(w_{\ell}^T H_{\ell} )) \approx 0$. However, if without residual connections, finding $w_{\ell}$ and $w_{\ell+1}$ such that $H_{\ell+2}=a(w_{\ell+1}^T a(w_{\ell}^T H_{\ell} )) \approx H_{\ell}$ might be difficult. The property that $H_{\ell+2} \approx H_{\ell}$ is more likely to be true when a network is deep. However, when using Delog-SCL or AL, each local network is short, so Delog-SCL and AL are unlikely to take advantage of the residual connections. As a result, optimizing a ResNet by BP usually gives better results than by the BP-decoupling methods, such as Delog-SCL and AL.
> > > >
> > > > All in all, no technique fundamentally solves the issue of vanishing/exploding gradients; they only mitigate the issue. Our work tackles the problem from a different perspective -- shorten the gradient flows. Therefore, the new design deserves to be discussed in ICLR.

---

### Official Review · Reviewer_tuDH · 2022-10-26

**Confidence:** 3
**Correctness:** 3
**Technical Novelty And Significance:** 3
**Empirical Novelty And Significance:** 2
**Recommendation:** 6

**Clarity, Quality, Novelty And Reproducibility:**

In general, the paper is well written. The motivations are clearly explained. The experimental design is also sound. One novelty of this paper is that it brings supervised contrastive learning to solve the problem of long gradient flow, complementing some existing approaches such work such as target propagation and AL. The source code is not provided for reproducibility.

Some minor issues and typos:

1. Figure 1 notations are a little bit confusing.

2. Page 4, line 7 after equation (1): $]r_1^{(i)}$.


**Strength And Weaknesses:**

This work has several strengths:

1. The proposed method is able to cut the long gradient flow in training deep neural networks.

2. The proposed method decouples the parameter updating across layers, making parallel training possible, though it needs follow-up work to validate this claim.

3. The proposed method requires fewer parameter than similar method such as AL in training phase, and has the same effective parameter in inference phase.

4. The test accuracy on simpler tasks and simpler architectures are better than AL and comparable to BP.

Weaknesses

1. One important aspect of learning procedures are the learning dynamics, which is not thoroughly studied. What are the learning curves like for the methods compared? How much longer or shorter does the proposed method converge compared to AL and BP? Is the training stable? Do we need to use different learning steps for each layer? How does the parameters change during training and use this to validate whether it actually addresses some problems with long gradient flows, such as the exploding/vanishing gradient problem for long gradient flows?

The research community would also benefit from some potential follow-up work such as the pipelining implementation and its possible extension to training other architectures such as RNNs.


**Summary Of The Paper:**

In this paper, the author proposed a supervised contrastive learning (SCL)-based training method for deep neural networks in which the gradients only flow locally in each layer, solving some of the problems associated with long gradient flows in backpropagation. The authors also benchmarked its performance against early exit, associated learning (AL) and backpropagation (BP) for training some standard feedforward network architectures for classification tasks. The results showed that the proposed method outperforms early exit, AL on most tasks while BP is still superior for ResNet.

In addition, the author also did some analysis on the effects of batch size on the final test performance and found larger batch size generally works better. The author also found that a nonlinear projection head performs better than identity or linear heads.


**Summary Of The Review:**

The authors proposed a supervised contrastive learning inspired approach to train deep neural networks without long gradient flows. The proposed method has several characteristics such as 1) it only requires local gradient computation and parameter updates 2) it enables parallel training via pipelining 3) it requires fewer additional parameters than similar methods in training phase and same number of parameters as BP in inference phase 4) the test accuracy are comparable to BP. The authors also studied the effects of batch size, non-linear projection head. However, a thorough study and comparison of the learning dynamics are missing.

---

> ### Author Response · Authors · 2022-11-18
> **Thank you for your insightful review and advices!**
>
> **Q1: What are the learning dynamics of Delog-SCL?**
>
> Delog-SCL is stable in training.  Figure 5 in the Appendix compares Delog-SCL and BP on vanilla CNN in terms of their dynamics of test accuracy on CIFAR-100 when the epoch increases.  First, the test accuracy of Delog-SCL improves stably.  Second, the Delog-SCL outperforms BP after approximately 100 epochs.
>
> Different hyperparameter settings may lead to slightly different curves.  However, most of them follow a similar pattern.  Experiments on other datasets (CIFAR-10 and tiny-ImageNet) for the VGG network structure also show similar trends.
>
> We added the above discussion in the main context (Sec.  3.1) and the Appendix (A.1).
>
> **Q2: The source code is not provided for reproducibility.**
>
> We uploaded the source code.  In addition, we included pseudo-code in Appendix A.5.
>
> **Q3: Figure 1 notations are a little bit confusing.**
>
> We updated Figure 1 and the description in Section 2.1 for clarification.
>
> **Q4: Page 4, line 7 after equation (1): ]r1(i).**
>
> Corrected.

---

> > ### Comment · Reviewer_tuDH · 2022-11-21
> > **Good improvement over the first draft**
> >
> > Thank you, authors, for your added experiments and updated manuscript. Glad to see the improvements and I would keep my recommendation as "marginally above the acceptance threshold".

---

### Author Response · Authors · 2022-11-18
**General Response to all reviewers**

Vanishing/exploding gradients are still fundamental issues in training a neural network, especially when a network is deep.  Many designs (e.g., ResNet and ReLU activation) attempt to address the issue, but none of them essentially solve it.  We approach the problem from another perspective: we cut a long end-to-end long gradient flow into smaller ones to alleviate vanishing/exploding gradients caused by a long gradient flow.   Our design (Delog-SCL) is not an alternative to ResNet or ReLU; they can be applied altogether to help pass gradients.

Separating a long gradient flow into smaller pieces and maintaining competitive test accuracy is challenging because local objectives may mismatch the global objective.  We compared Delog-SCL with AL (the state-of-the-art method to separate long gradient flows) and Early Exit (a classic technique to separate long gradient flows).  Delog-SCL usually performs better than AL and requires much minor modification on the structure of an existing network.  As a promising alternative to end-to-end backpropagation, this work should be discussed in ICLR.

---

### Author Response · Authors · 2022-11-18
**Manuscript updated**

Dear reviewers,

Thank you for your great suggestions. We have modified our manuscript as suggested.

1. We added experiments to confirm cutting a long gradient flow into multiple short ones accelerates learning (**Section 3.3**).
1. We uploaded the experimental code for reproducibility. (**[Supplementary material](https://openreview.net/attachment?id=6BO4lP8K1N1&name=supplementary_material)**)
1. We added a comparison of training and testing time of BP, AL, and Delog-SCL (**Appendix A.2**).
1. We provided pseudo code for constructing and training Delog-SCL (**Appendix A.5**).
1. We discussed why Delog-SCL and AL could not take advantage of skip connections  (**Section 3.1.1**).
1. We added the relationship between test accuracy and epoch for Delog-SCL to confirm the learning is stable (**Appendix A.1**).

---

### Comment · Area_Chair_uFxq · 2022-11-28
**Reviewers please respond**

Dear reviewers,

the authors have tried to address your points in your reviews. It would be cordial to the authors, and helpful to me, if you could clarify if the rebuttal is to your satisfaction, or if there are remaining points to be addressed. Otherwise, there is little point to a rebuttal.

With kind regards,
Your AC

---

### Decision · Program_Chairs · 2023-01-20

**Decision:**

Reject

**Justification For Why Not Higher Score:**

The novelty and contributions is not clear, and at the same time Delog-SCL does not consistently outperform AL across all models and datasets. There are some elements in the ablation that are substantial, e.g. Vanilla ConvNet on Cifar100. Upon further inspection, one finds that the "Vanilla ConvNet" is a 3-layer Conv Net that I assume has been hand-engineered by the authors to outperform on the task. All in all, either the theoretical contributions must be further boosted, or the empirical findings become more generalizable across settings/datasets and detailed.

**Justification For Why Not Lower Score:**

The paper is generally on a good track, I expect with further imrpovements it can be accepted in a good conference.

**Metareview: Summary, Strengths And Weaknesses:**

This paper proposes an algorithm that breaks long gradient flows in end-to-end backpropagation to multiple short ones, as the objective is the inefficiencies of regular backpropagation when going with deeper networks. The main idea of this algorithm is to use of supervised contrastive learning, where examples of the same category should be close in the embedding space. Specifically, in the backward path when computing gradients, the flow is stopped per components in the network, and a 'global' objective is replaced my multiple local ones. Effectively the proposed algorithm considers deep networks as cascaded shallow ones, where each intermediate layer is not 'hidden' per se; rather, it considers the input from the previous layer as an observation variable (so, not a learned variable), and returns an output embedding that can be evaluated with a supervised contrastive loss. A problem that this approach can help with is vanishing gradients, which makes sense since intermediate loss functions are introduced into the neural network. This is not surprising, as this was also used with Inception networks, of course with regular backpropagation so that to be able to train deeper networks. For what is worth, the abstract is not very clear on what is the algorithm supposed to do, and only explains that a 'novel methodology' is proposed.

This was a paper with both positive and negative ratings. All reviewers agreed that the importance of the theoretical contributions of the paper is not clear, and I agree with that statement. The difference with Greedy Infomax is not persuasive enough. The authors argue that augmentations are only possible with visual tasks, however, this argument is not entirely valid (eg a simple Google Search for 'Molecular Contrastive Learning ...' returns quite a few results). What is more, replacing the requirement for augmentations with an arguably an even more expensive one (labelled samples), is not that convincing.

Besides, introducing loss layers in every layer in between is not novel itself and it was introduced for a similar observation of vanishing gradients (see Inception family of architectures). It was not considered as an option of non-end-to-end training, however, I think this is a rather minor difference. In other words, I do not this it is fair to say that said algorithm breaks the long gradient flow to smaller ones, because the gradient is not the same (and not supposed to be the same), as the learning objective changes massively. And if the intent was the gradients to be similar enough, then likely there is substantial bias added to the gradient estimation, and an ablation and theoretical study would be warranted.